# All-Solid-State Potentiometric Platforms Modified with a Multi-Walled Carbon Nanotubes for Fluoxetine Determination

**DOI:** 10.3390/membranes12050446

**Published:** 2022-04-21

**Authors:** Hisham S. M. Abd-Rabboh, Heba M. Hashem, Layla M. S. Al Shagri, Abdel El-Galil E. Amr, Abdulrahman A. Almehizia, Ahmed M. Naglah, Ayman H. Kamel

**Affiliations:** 1Chemistry Department, Faculty of Science, King Khalid University, P.O. Box 9004, Abha 61413, Saudi Arabia; hasalah@hotmail.com; 2Department of Chemistry, Faculty of Science, Ain Shams University, Cairo 11566, Egypt; hebahashem426@yahoo.com; 3Chemistry Department, College of Science, Sakhir 32038, Bahrain; lalshagri@uob.edu.bh; 4Pharmaceutical Chemistry Department, College of Pharmacy, King Saud University, Riyadh 11451, Saudi Arabia; mehizia@ksu.edu.sa (A.A.A.); anaglah@ksu.edu.sa (A.M.N.)

**Keywords:** screen printed, potentiometric sensors, multi-walled carbon nanotubes (MWCNTs), fluoxetine, nanomaterials-based sensors

## Abstract

Novel cost-effective screen-printed potentiometric platforms for simple, fast, and accurate assessment of Fluoxetine (FLX) were designed and characterized. The potentiometric platforms integrate both the FLX sensor and the reference Ag/AgCl electrode. The sensors were based on the use of 4′-nitrobenzo-15-crown-5 (ionophore I), dibenzo-18-crown-6 (ionophore II), and 2-hydroxypropyl-β-cyclodextrin (2-HP-β-CD) (ionophore III) as neutral carriers within a plasticized PVC matrix. Multiwalled carbon nanotubes (MWCNTs) were used as a lipophilic ion-to-electron transducing material and sodium tetrakis [3,5-bis(trifluoromethyl)phenyl] borate (NaTFPB) was used as an anionic excluder. The presented platforms revealed near-Nernstian potentiometric response with slopes of 56.2 ± 0.8, 56.3 ± 1.7 and 64.4 ± 0.2 mV/decade and detection limits of 5.2 × 10^−6^, 4.7 × 10^−6^ and 2.0 × 10^−7^ M in 10 mM Tris buffer solution, pH 7 for sensors based on ionophore I, II, and III, respectively. All measurements were carried out in 10 mM tris buffer solution at pH 7.0. The interfacial capacitance before and after insertion of the MWCNTs layer was evaluated for the presented sensors using the reverse-current chronopotentiometry. The sensors were introduced for successful determination of FLX drug in different pharmaceutical dosage forms. The results were compared with those obtained by the standard HPLC method. Recovery values were calculated after spiking fixed concentrations of FLX in different serum samples. The presented platforms can be potentially manufacturable at large scales and provide a portable, rapid, disposable, and cost-effective analytical tool for measuring FLX.

## 1. Introduction

Fluoxetine (FLX), *N*-methyl-3-phenyl-3-[4-(trifluoromethyl) phenoxy] propan-1-amine, is a medical drug primarily used to manage major depressive disorder (MDD). MDD is characterized by depressed mood, loss of interest in daily activities, altered cognitive function, and deterioration in physical health, resulting in a reduced quality of life [1]. FLX is used as a drug therapy for MDD treatment as it lifts mood without major side effects and prevents disease relapse [2]. Both R and S enantiomers racemate in equimolar amount and classified as an inhibitor of selective serotonin reuptake (SSRI) [3]. The importance of FLX as a selective serotonin reuptake inhibitor (SSRI) is that it is safer than other antidepressants that have adverse effects and is therefore approved for pregnant women and adolescents, as well as children [4]. FLX is metabolized by N- de-methylation to the active metabolite norfluoxetine in the liver. However, its production takes a few days because FLX has a longer half-life than its metabolite norfluoxetine [5]. So, it has become difficult to measure the active metabolite in plasma. Longer duration studies are required, which adds to the limitations of such studies involving human volunteers. As a result, most analytical studies are based on measuring the levels of FLX and not its metabolite in biological fluids to extrapolate the pharmacokinetics and pharmacodynamics [6].

Different analytical methods were reported for FLX assessment in both pharmaceutical dosage and biological fluid samples, including the high-pressure liquid chromatography (HPLC) [7,8,9], gas chromatography, [10,11,12] liquid-chromatography coupled with MS detection, [13,14,15] capillary electrophoresis, [16,17,18] fluorimetry, [19] spectrometry, [20,21,22,23] voltammetry, [24,25,26,27] and potentiometry [28,29,30,31]. Almost all these methods are sophisticated, imply high-cost instruments, have long run-time for analysis and require well-trained analysts. In addition, they imply manual extraction from the biological samples, followed by subsequent chromatographic separation and quantification. This manual extraction step results in these methods having low sensitivity, narrow range, and requiring a large volume of biological samples.

Among these analytical methods, all-solid-state ion-selective electrodes (ISEs) based on potentiometric transductions revealed several merits. They are cost-effective, rapid, accessible, and precise analysis, simple instrumentation, and incorporated functionality. In addition, they offer a practical viable method without sample pre-treatment, prolonged analysis time and sophisticated experimental establishment [32,33,34,35]. All solid-state screen-printed ISEs have been chosen for flexible, reliable, and low-cost platforms for potentiometric analytical devices. [36,37,38,39] The advantages and limitations of the previously reported potentiometric sensors [28,29,30,31] in comparison with the presented sensors are shown in Table 1.

In this work, a reliable, robust, simple, and cost-effective analytical method based on potentiometric detection for FLX assessment was presented. The method is based on the preparation and characterization of new potentiometric all solid-state screen-printed planar electrodes. Three neutral carrier ionophores were used as artificial receptors for the recognition of fluoxetine. MWCNTs were used as an ion-to-electron transducer. The performance characteristics in terms of detection limit, linearity, sensitivity, selectivity, accuracy, intra-day and inter-day repeatability, potential stability, method robustness, and method ruggedness were tested and evaluated. Successful applicability of the presented sensors was carried out for FLX determination in different pharmaceutical preparation samples. Method recovery was evaluated using spiking addition method after spiking known amounts of FLX in different human serum samples.

## 2. Materials and Methods

### 2.1. Chemicals and Reagents

High molecular weight poly (vinyl chloride) (PVC), dioctyl phthalate (DOP), *o*-nitrophenyloctyl ether (o-NPOE), dibutyl sebacate (DBS), 2-hydroxypropyl-β-cyclodextrin (2-HP-β-CD), sodium tetrakis [3,5-bis(trifluoromethyl)phenyl] borate (NaTFPB), tris(hydroxymethyl)aminomethane (Tris), 4′-nitrobenzo-15-crown-5 (ionophore I) and dibenzo-18-crown-6 (ionophore II), were obtained from Sigma Aldrich (St. Louis, Missouri, MO, USA). Tetrahydrofuran (THF) was obtained from Fluka AG (Buchs, Switzerland) and freshly distilled prior to use. Ag/AgCl ink (E2414) was purchased from Ercon (Wareham, MA, USA). MWCNTs were purchased from (EPRI, Cairo, Egypt). Pure fluoxetine.HCl (FLX) was obtained from Pharaonia Pharmaceuticals (Alexandria, Egypt). All drugs containing FLX were collected from the local market such as Prozac (20 mg/capsule; Lilly, France), Philozac (20 mg/capsule; Amoun, Egypt), Flutin (20 mg/capsule; Eipico, Egypt) and Depreban (20 mg/capsule; Amirya, Egypt). 

For the preparation of the stock FLX solution (10^−2^ M), a definite weight of the pure drug was dissolved in 100 mL de-ionized water. The working solutions (10^−7^–10^−2^ M) were prepared after accurate dilution of the stock solution. The solutions were stored in brown bottles and kept in the refrigerator. All calibration measurements were carried out in 10 mM Tris buffer solution of pH 7.

### 2.2. Apparatus

A Millipore Milli-Q system was used for obtaining de-ionized water (18.2 MΩ·cm specific resistance) to prepare all solutions. All potentiometric experiments were carried out at 25 ± 1 °C, using pH/mV meter (PXSJ-216, INESA-Scientific Instrument Co., Ltd., Shanghai, China). Chronopotentiometric measurements were carried out using a three-electrode cell containing the screen-printed electrode and a platinum wire as an auxiliary electrode. For these measurements, Metrohompotentiostat/galvanostat (Autolab-model 204, Herisau, Switzerland) was used.

### 2.3. Electrode Fabrication and Potential Measurements

The screen-printed electrode (SPE) was made from ceramic as a supporting substrate with 0.1 mm thickness and 35 mm length. Two screens were printed and coated with carbon ink. For the preparation of the reference electrode, one of the carbon screens was coated with Ag/AgCl ink and then dried at 70 °C for 10 min. The reference membrane cocktail was prepared by dissolving 78.1 mg polyvinyl butyral (PVB), 50 mg NaCl in 1 mL methanol. 10 µL of this solution was drop-casted on the Ag/AgCl orifice and left to dry overnight. MWCNTs were dissolved in THF (1 mg/mL), and 15 µL was deposited by drop-casting onto the carbon sensing area. After drop-casting, the solution was left to dry for 3 min. The FLX-selective membrane contained 100 mg of the components in 1.5 mL THF as: Either ionophore I, II or III (2.0 %), KTFPB (1.0%), PVC (32.0%), and plasticizer (65%). A 15 µL of the membrane solution was added by drop-casting onto the modified carbon orifice and left to dry overnight. The prepared screen-printed electrodes were conditioned before use in a mixture solution containing 10^−3^ M FLX.HCl (pH 7.0) and 1 M NaCl for 4 h.

The electromotive forces (emf) were measured at 25 ± 1 °C. The constructed potentiometric cell was immersed in stirred solutions. A correction was made for the EMF values according to the Henderson equation to eliminate the liquid-junction potential. Activity coefficients of the working standard FLX solutions were evaluated according to Debye–Huckel approximation. The electrode’s performance characteristics were evaluated according to the IUPAC recommendations [40,41].

### 2.4. Chronopotentiometric Measurements

Constant-current chronopotentiometry measurements were conducted to test the short-term potential stability of the presented electrodes and to evaluate the double-layer capacitance of both MWCNTs as an ion-to electron transducer [42]. The designed electrochemical cell was connected in a one-compartment cell in 10 mM FLX at room temperature. The auxiliary electrode was a Pt wire. A constant current of ±1 nA was applied on the working electrode for 60 s followed by a reversed current for another 60 s.

### 2.5. Application to Real Samples

Four commercially available drugs containing FLX were chosen to test the applicability of the presented electrodes. They were represented commercially as Prozac, Philozac, Flutin, and Depreban capsules. All of these samples contain 20 mg FLX/capsule. Five capsules from each drug type were grinded and accurately weighed. The weighed amount of the powder was dissolved in 10 mM Tris buffer solution, pH 7 and sonicated until complete dissolution for 45 min. The solution is completed by the buffer to 100 mL resulting in 1000 µg/mL FLX stock solution. Different concentrations were prepared after dilution from the stock solution. From the constructed calibration plot, the amount of FLX in the samples was determined under the same conditions.

To test the applicability of the presented sensors towards FLX determination in complicated matrices, different human serum samples were spiked with known amounts of FLX. A total of 9-mL (100 µL of human serum + 8.9 mL of 10 mM Tris buffer, pH 7) was placed in a 20-mL beaker. An aliquot of different FLX concentration solutions (1.0 mL) was added to the human sample and thoroughly mixed and used for FLX measurements. The analytical device was then inserted in the test solution and the potential readings were recorded after stabilization. From the calibration plot, the FLX concentration was calculated.

## 3. Results

### 3.1. Sensors’ Characterizations

All solid-state ion-selective sensors responsive for FLX were designed, characterized, and successfully applied for the drug analysis. The membrane sensors were based on 4′-nitrobenzo-15-crown-5 (ionophore I), dibenzo-18-crown-6 (ionophore II), and 2-hydroxypropyl-β-cyclodextrin (2-HP-β-CD) (ionophore III) as recognition receptors. MWCNTs were used as solid-contact transducers between the ion-sensing membrane and the electrode conductor. For membrane optimization, three different plasticizers were used to test their polarities’ effect on the sensitivity and selectivity of the presented potentiometric sensors. The performance characteristics of the presented sensors were electrochemically evaluated, and the results were listed in Table 2.

The obtained results showed that all ionophores can form an inclusion complex with hydrophobic guest molecules, because their cavities are exo-hydrophilic endo-hydrophobic [43]. They act as neutral carrier incorporating strong multiple hydrogen bond donor groups (-O-) which assist conformational adjustments of FLX for maximum Vander Waals forces [44]. The size of ionophore III cavity fits and accommodates the FLX molecule more than ionophores I and II.

For sensors based on ionophore I, they revealed a near-Nernstian response towards FLX ions with slopes of 56.2 ± 0.8, 40.5 ± 1.2 and 40.4 ± 0.7 mV/decade and detection limits of 5.2 × 10^−6^, 6.0 × 10^−6^ and 1.0 × 10^−5^ M for the membranes plasticized with o-NPOE, DOP, and DBS, respectively. Sensors based on ionophore II exhibited near-Nernstian slopes of 56.3 ± 1.7, 52.2 ± 1.4 and 53.6 ± 0.3 mV/decade with detection limits of 4.7 × 10^−6^, 6.3 × 10^−6^ and 2.0 × 10^−5^ M for membranes plasticized with o-NPOE, DOP, and DBS, respectively. For sensors based on ionophore III, they exhibited near-Nernstian potentiometric response with slopes of 64.4 ± 0.2, 48.4 ± 0.7 and 43 ± 0.4 mV/decade and detection limits of 2.0 × 10^−7^, 8.0 × 10^−7^ and 3.2 × 10^−6^ M for the membranes plasticized with o-NPOE, DOP, and DBS, respectively.

The results obtained revealed that the solvent polarity of the membrane plasticizer significantly affects the response of the sensors. Figure 1A,B showed that high dielectric constant plasticizer (e.g., o-NPOE, є = 24) is more favorable than low-dielectric constant plasticizers (e.g., DOP, є = 7 and DBS = 4) [45].

The pH effect on the potential response of the presented electrodes was examined using 1.0 × 10^−5^ and 1.0 × 10^−4^ M FLX solutions over different pH values starting from pH 2 to pH 10. Solution adjustment was carried out using either LiOH or HCl. As shown in Figure 2, the pH-potential profiles showed that FLX membrane sensors revealed good potential stability over the pH range 4.5–8.5 and 4–9 for sensors based on ionophore I, ionophore II and ionophore III, respectively. At pH < 4, an increase in the potential was observed due to interference from the high H^+^ concentration. At pH > 9, a noticeable potential decrease is observed due to the formation of the free basic drug. The response time was evaluated as the time required attaining a stable potential after increasing the FLX concentration and was typically <5 s especially for low FLX concentrations (Figure 1). 

### 3.2. Selectivity Behavior

The selectivity behavior of the fabricated sensors was evaluated using the fixed interference method using fixed concentration (10^−2^ M) of the interfering ion [46] In Table 3, it summarizes the selectivity coefficient values for sensors based on ionophores I and II in different plasticizers. The selectivity coefficient values depend on the composition of the membrane and varied widely from one type to another [47]. Under the optimum conditions previously mentioned, these values were evaluated and calculated. The selectivity order for ionophore I membrane-based sensor was: FLX > norfluoxetine > K^+^ > Na^+^ > Rb^+^ > lactose > Zn^2+^ > glucose > Li^+^ > caffeine > paracetamol > arginine. The selectivity pattern of ionophore II membrane-based sensor was: FLX > norfluoxetine > K^+^ > Rb^+^ > Na^+^ > lactose > paracetamol > caffeine > arginine > Zn^2+^~glucose > Li^+^ > Ca^2+^ > Ba^2+^. For sensor III, the selectivity order was: FLX > norfluoxetine > K^+^ > lactose > Rb^+^ > glucose > Na^+^ > Ca^2+^ > Ba^2+^ > Zn^2+^ > Li^+^ > paracetamol > caffeine > arginine. From the selectivity coefficient values presented in Table 3, it was found that Ionophore III exhibited better selectivity behavior than ionophores I and II, especially for the metabolite norfluoxtine. The response mechanism of the presented ionophores towards FLX cation is based on the ion-complex properties between FLX cations and the configuration structure and cavity size of the crown ether ionophore in the polymer matrix. The electrostatic interaction between FLX cation and ether group in the crown ether ionophore plays the dominant role for the cation transfer across the organic/water interface. The electrostatic affinity is overcome by the hydration energy of the analyte cations. So, the obtained selectivity sequence is determined by the hydrophilicity of the tested cations rather than the order of their hydration energies. 

The cavity radius of either 15-crown-5 and 18-crown-6 are 0.85–1.1 and 1.3–1.6 A°, respectively [48]. The inclusion of FLX with macrocyclic compounds requires a bigger cavity size. This reflects the better selectivity of ionophore II than ionophore I. For β-CD compounds, they revealed cavity sizes in the range of 6.0–6.5 A°. It is well reported related in the literature, the high affinity of the aromatic groups to accommodate in the cyclodextrin cavity, favoring the van der Waals interactions [49]. In this sense, the enthalpy changes could be attributed to the binding of enthalpy-rich water molecules, released from the β-CD cavity, with bulk water molecules, as well as the formation of cooperative van der Waals interactions between guest host, and, mainly, the electrostatic interaction between FLX and unpaired electrons of OH groups, explaining the higher enthalpic contribution.

### 3.3. Repeatability, Reproducibility, and Stability

Repeatability, reproducibility, and stability of an FLX-based sensor were checked using potentiometric measurements of standard FLX solution (10.0 µM, *n* = 6). The relative standard deviation (RSD%) for measuring this concentration was found to be 2.1% and 2.2% for sensors based on ionophores I and II, respectively. This can be considered as adequate repeatability. Reproducibility was measured after measuring the above-mentioned concentration using different sensor assembly and different instruments at different times. The sensors revealed good reproducibility with an RSD% (*n* = 6) of 2.3%. 

The lifespan of the sensors was also tested and shown in Figure 3. During 10 days-working, the sensors revealed an acceptable response from their initial response. This indicates that the proposed platform was excellent and enabled good stability.

### 3.4. Water-Layer Test

The water-layer test was performed to evaluate the lipophilicity of the solid-contact transducing material and the ability of the sensor to exclude water from the contact between the electronic conducting substrate and the sensing-ion membrane [50].

Both the modified and non-modified platforms for ionophores I and II membrane-based sensors were sequentially immersed in 0.1 M NaCl, 0.1 mM FLX, and 0.1 M NaCl solutions. As shown in Figure 4, all modified platforms revealed a stable potential-response during the test. There are no long-term drifts in the potential on switching from one solution to the other.

This demonstrates the high lipophilicity of MWCNTs layer and confirms the non-existence of the water layer at interface between the ion-sensing membrane and the electronic substrate. It was necessary to demonstrate that FLX sensors were free of a detrimental water layer because water layer is crucial in obtaining low detection limits for potentiometric sensors. The composition of the ultrathin water layer is altered due to the ion-exchanging on the inner side of the membrane. This leads to drifts in the backside solid-contact sensor’s potential [51].

### 3.5. Short-Term Potential Stability and Interfacial Capacitances

Short-term potential stability for the presented platforms and interfacial capacitances in absence and presence of the solid-contact transducing material were evaluated by applying the reverse-current chronopotentiometry method presented by Bobacka [42].

The applied current (*I*) was ±1 nA. The chronopotentiograms for both sensor I and sensor II in presence of MWCNTs, together with sensor I and sensor II in absence of MWCNTs, were shown in Figure 5. The potential drifts (Δ*E*/Δ*t*) were calculated for sensors I, II, and III in the presence and absence of MWCNTs and tabulated in Table 4.

The double layer capacitances arisen from the insertion of the solid-contact transducer [*C_L_* = *I*/(Δ*E*/Δ*t*)] were evaluated for all presented sensors and tabulated in Table 4. 

The results confirmed that insertion of MWCNTs between the electronic conductor substrate and ion-sensing membrane revealed high potential stability of the sensors and reflects the well-confined ion- to electron transduction process. Unmodified sensors that have no MWCNTs exhibited low-double-layer capacitances that revealed low potential stability. Therefore, this electrode is seen to be polarizable without the ability to buffer any random tiny charge noise.

### 3.6. Analytical Applications

The presented platforms were successfully applied to quantify the amount of FLX in different pharmaceutical dosage forms. Construction of standard calibration curve using pure FLX prepared in 10 mM Tris buffer solution of pH 7 was used for the drug assay. Table 5, Table 6 and Table 7 showed that the data analysis for different FLX samples (five replicate measurements) was acceptable, which confirms the applicability of the presented platforms for FLX determination. The obtained potentiometric results were compared with the standard liquid chromatographic method (HPLC) [52]. The *t*-student and *F*-tests were calculated for the two methods. They showed no significant difference, which confirmed the successful application of the presented potentiometric method. The method showed high efficiency in FLX determination in different matrices.

To test the applicability of the presented platforms in medical applications, FLX was spiked and determined in different human blood serum samples. The average recoveries were found to be 98.9, 98.1, and 98.2% with a relative standard deviation of ±0.7%, ±1.1% and 0.4% for sensors I, II, and III, respectively. All obtained results for FLX assessment in these spiked human serum samples were shown in Table 8.

## 4. Conclusions

In summary, novel FLX screen-printed sensors based on 4′-nitrobenzo-15-crown-5 (ionophore I), dibenzo-18-crown-6 (ionophore II) and 2-hydroxypropyl-β-cyclodextrin (2-HP-β-CD) (ionophore III) for potentiometric sensing of Fluoxetine (FLX) were fabricated, characterized, and presented. The platforms were modified by multi-walled carbon nanotubes (MWCNTs) as lipophilic nanomaterial and ion-to-electron transducer. The sensors revealed a Nernstian potentiometric response with slopes of 56.2 ± 0.8, 56.3 ± 1.7 and 64.4 ± 0.2 mV/decade and detection limits of 5.2 × 10^−6^, 4.7 × 10^−6^ and 2.0 × 10^−7^ M in 10 mM Tris buffer solution, pH 7 for sensors based on ionophore I, II, and III, respectively. The effect of solvent polarity on the potentiometric response and selectivity behavior of the sensors was studied. Several prominent merits were possessed for the presented sensors, such as high potential-stability, eco-friendly property, fast response, good recognition specificity, and enhanced repeatability and reproducibility. The obtained good performance characteristics confirmed successful applicability for the accurate and quick determination of FLX in pharmaceutical formulations and human serum samples. This work can be directed to further low-cost and disposable screen-printed based analytical devices for potentiometric sensing produced at large scales with high speed and reproducible screen-printing technology.

## Figures and Tables

**Figure 1 membranes-12-00446-f001:**
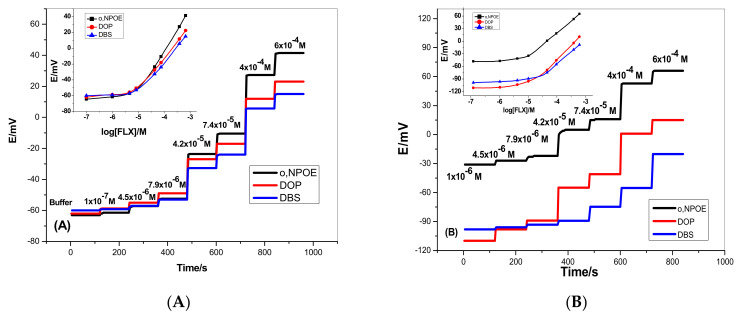
Time-trace versus FLX concentration for sensors based on (**A**) ionophore I (**B**) ionophore II; (**C**) ionophore III; using o-NPOE, DOP and DBS as membrane solvent mediators. (Inset: calibration plot).

**Figure 2 membranes-12-00446-f002:**
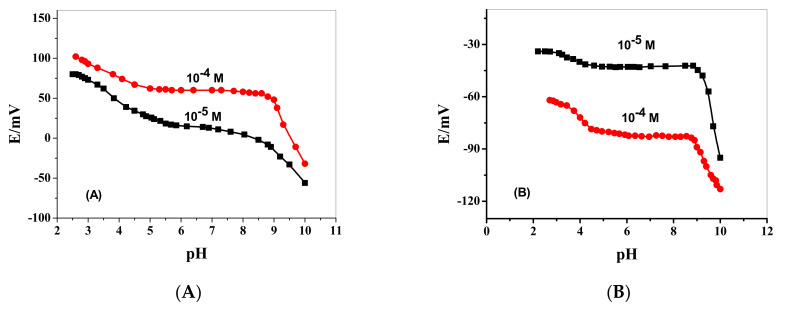
pH-potential profiles for FLX membrane sensors plasticized in o, NPOE (**A**) ionophore I; (**B**) ionophore II and (**C**) ionophore III.

**Figure 3 membranes-12-00446-f003:**
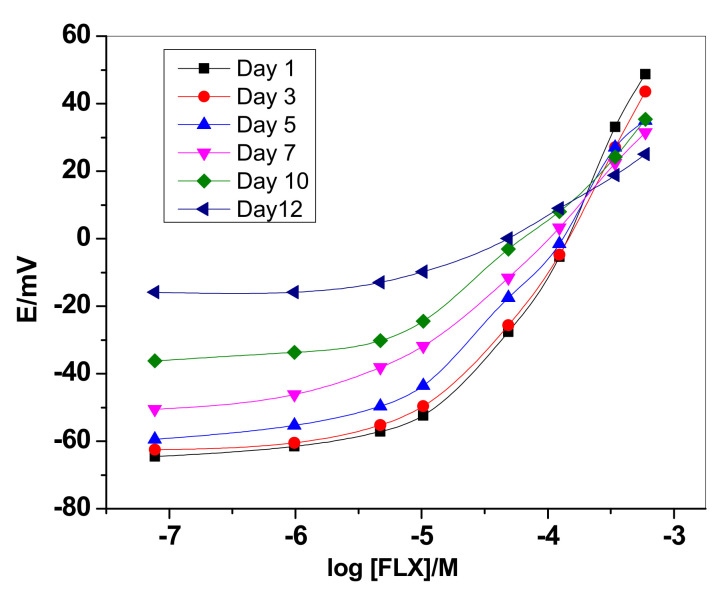
Day-to-day performance characteristics of ionophore III based sensor.

**Figure 4 membranes-12-00446-f004:**
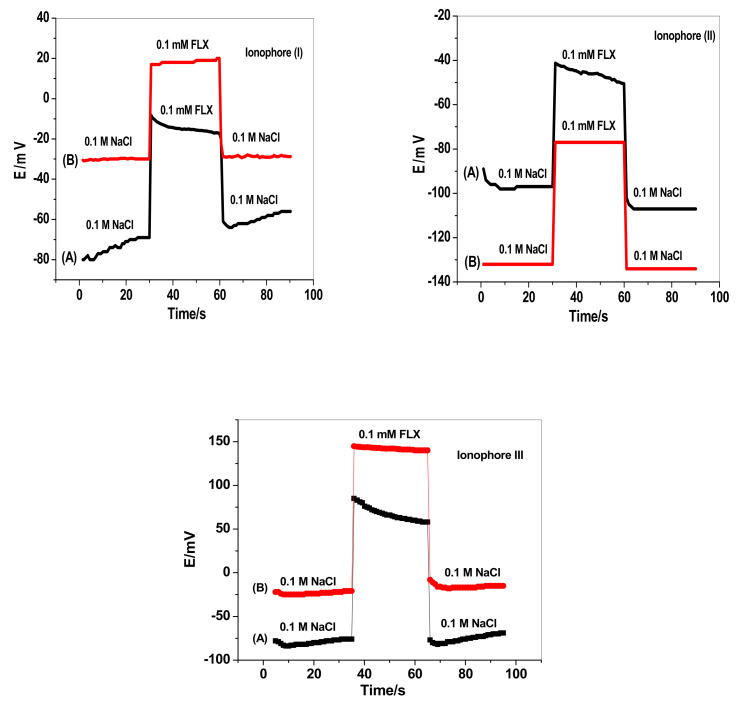
Water layer tests for (**A**) non-modified, (**B**) and modified FLX sensors based on ionophores I, II, and III.

**Figure 5 membranes-12-00446-f005:**
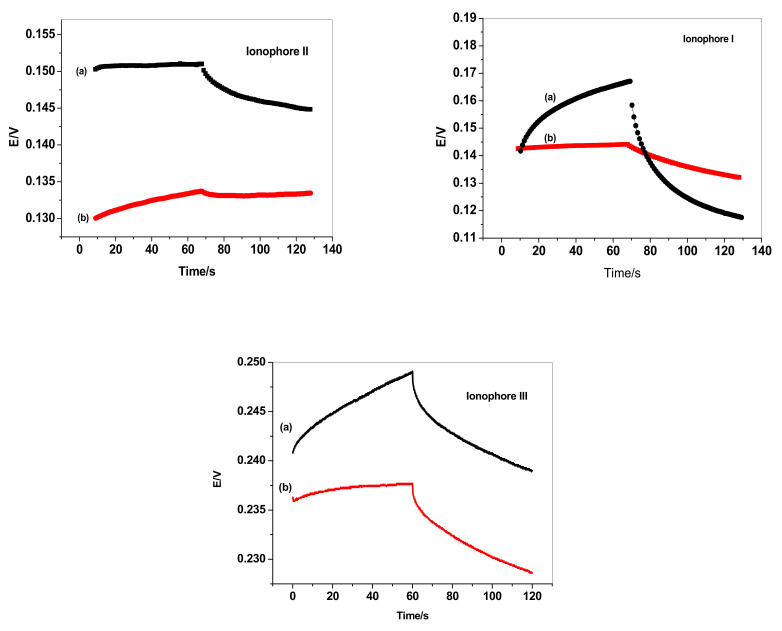
Current reversal chronopotentiometry for (**a**) non-modified and (**b**) modified FLX-ISEs based on ionophores I, II, and III.

**Table 1 membranes-12-00446-t001:** Comparison of the presented sensors with previously reported potentiometric ion-selective electrodes (ISEs).

Sensing Material	Electrode Type	Slope, mV/DECADE	Detection Limit, M	Lower Limit of Linear Range, M	Working pH Range	Ref.
Fluoxetine/picrolonate	Liquid polymeric	51 ± 0.5	6 × 10^−6^	8 × 10^−6^	1–5	28
Fluoxetine/tetraphenylborate	Liquid polymeric	58.5	2.3 × 10^−5^	4.3 × 10^−5^	4.0–7.5	29
	Coated wire graphite electrode	55.5	2.5 × 10^−5^	5.4 × 10^−5^		
Fluoxetine/tetraphenylborate	Liquid polymeric	51	3.0 × 10^−6^	3.0 × 10^−6^	4.0–7.5	30
Fluoxetine/phosphotungstate		51.8	5.0 × 10^−6^	5.0 × 10^−6^		
Molecular imprinting polymer (MIP), acrylamide	Solid-contact ISEs	58.9 ± 0.2	2.1 × 10^−6^	1.0 × 10^−5^	10 mM acetate buffer of pH 4.5	31
Ionophore I	Solid-contact ISEs	56.2 ± 0.8	5.2 × 10^−6^	6.5 × 10^−6^	10 mM Tris buffer solution of pH 7	This work
Ionophore II		56.3 ± 1.7	4.7 × 10^−6^	5.6 × 10^−6^		
Ionophore III		64.4 ± 0.2	2.0 × 10^−7^	2.0 × 10^−7^		

**Table 2 membranes-12-00446-t002:** Potentiometric response characteristic of FLX sensors in 10 mM Tris buffer, pH 7.

Parameters	Ionophore I			Ionophore II			Ionophore III		
	o-NPOE	DOP	DBS	o-NPOE	DOP	DBS	o-NPOE	DOP	DBS
Slope (mV/decade)	56.2 ± 0.8	40.5 ± 1.2	40.4 ± 0.7	56.3 ± 1.7	52.2 ± 1.4	53.6 ± 0.3	64.4 ± 0.2	48.4 ± 0.7	43 ± 0.4
Detection limit (M)	5.2 × 10^−6^	6.0 × 10^−6^	1.0 × 10^−5^	4.7 × 10^−6^	6.3 × 10^−6^	2.0 × 10^−5^	2.0 × 10^−7^	8.0 × 10^−7^	3.2 × 10^−6^
Correlation coefficient (R^2^)	0.999	0.999	0.997	0.999	0.999	0.998	0.999	0.997	0.998
Linear range (M)	6.5 × 10^−6^–1.0 × 10^−2^	6.5 × 10^−6^–1.0 × 10^−2^	4.0 × 10^−5^–1.0 × 10^−2^	5.6 × 10^−6^–1.0 × 10^−2^	4.5 × 10^−6^–1.0 × 10^−2^	4.6 × 10^−5^–1.0 × 10^−2^	6.0 × 10^−7^–1.0 × 10^−2^	3.2 × 10^−6^–1.0 × 10^−2^	1.0 × 10^−5^–1.0 × 10^−2^
pH range (pH)	4.5–8.5	4.5–8.5	4.5–8.5	4–9	4–9	4–9	4–9	4–9	4–9
Precision (mV %)	1.2	1.4	1.5	1.3	1.5	1.2	1.3	0.9	1.1
Accuracy (mV %)	99.8	99.0	99.1	99.7	99.8	99.1	98.8	99.2	98.5
Standard deviation (mV)	0.5	0.4	0.6	0.5	0.8	0.9	0.4	0.3	0.7

**Table 3 membranes-12-00446-t003:** Selectivity coefficients of both ionophore I and II membrane-based sensors plasticized in o-NPOE.

Interfering Ion, J	*Log K^Pot^ _FLX, J_* + SD *		
	Ionophore I	Ionophore II	Ionophore III
Li^+^	−5.1 ± 0.1	−4.5 ± 0.1	−4.8 ± 0.3
Na^+^	−3.7 ± 0.1	−3.6 ± 0.2	−4.0 ± 0.2
K^+^	−2.9 ± 0.3	−2.3 ± 0.2	−3.1 ± 0.1
Rb^+^	−4.0 ± 0.1	−2.5 ± 0.1	−3.5 ± 0.4
Ca^2+^	−5.1 ± 0.1	−4.6 ± 0.1	−4.5 ± 0.2
Zn^2+^	−4.7 ± 0.2	−4.4 ± 0.3	−4.7 ± 0.1
Ba^2+^	−5.3 ± 0.2	−4.9 ± 0.2	−4.6 ± 0.2
Arginine	−5.8 ± 0.4	−4.3 ± 0.2	−5.6 ± 0.3
Caffeine	−5.2 ± 0.1	−4.2 ± 0.1	−5.1 ± 0.1
Glucose	−5.0 ± 0.1	−4.4 ± 0.2	−3.9 ± 0.3
Lactose	−4.6 ± 0.2	−3.7 ± 0.1	−3.3 ± 0.2
Paracetamol	−5.4 ± 0.3	−3.9 ± 0.2	−4.9 ± 0.2
Norfluoxetine	−1.0 ± 0.6	−0.7 ± 0.02	−1.5 ± 0.1

* Average of 3 measurements.

**Table 4 membranes-12-00446-t004:** Potential drifts and double-layer capacitances for the presented sensors in the presence and absence of MWCNTs.

	Ionophore I		Ionophore II		Ionophore III	
	Without MWCNTs	With MWCNTs	Without MWCNTs	With MWCNTs	Without MWCNTs	With MWCNTs
Potential drift (Δ*E*/Δ*t*), µV/s	815.3 ± 3.4	95.9 ± 1.1	88.9 ± 1.5	19.4 ± 1.1	120.5 ± 2.1	24.6 ± 1.4
*C_L_*, µF	1.2 ± 0.7	10.4 ± 0.2	11.2 ± 2.6	51.5 ± 2.6	8.29 ± 1.3	40.6 ± 2.1

**Table 5 membranes-12-00446-t005:** Determination of FLX in different pharmaceutical preparations using ionophore (I) membrane-based sensor.

Pharmaceutical Product and Source	Nominal Content Taken, mg/Tablet	Found, mg/Tablet				*t*-Student Test ^b^	*F*-Test
		Proposed Method	Mean ^a^ (%) ± SD	Reference Method, [47]	Mean ^a^ (%) ± SD		
Prozac (Lilly, France)	20	20.04	100.2 ± 0.4	20.1	100.8 ± 0.6	1.62	2.24
Philozac (Amoun, Egypt)	20	19.93	99.7 ± 0.6	19.8	99.07 ± 1.7	0.38	9.35
Flutin (Eipico, Egypt)	20	20.21	101.0 ± 1.4	19.8	99.4 ± 0.9	3.69	2.66
Depreban (Amirya, Egypt)	20	19.72	98.6 ± 0.8	19.4	97.2 ± 0.8	2.13	1.08

^a^ Mean of three replicates. ^b^
*t*-Student and *F*-test test at 95% confidence level values are 4.30 and 19.00, respectively.

**Table 6 membranes-12-00446-t006:** Determination of FLX in different pharmaceutical preparations using ionophore (II) membrane-based sensor.

Pharmaceutical Product and Source	Nominal Content Taken, mg/Tablet	Found, mg/Tablet				*t*-Student Test ^b^	*F*-Test
		Proposed Method	Mean ^a^ (%) ± SD	Reference Method	Mean ^a^ (%) ± SD		
Prozac (Lilly, France)	20	20.9	102.2 ± 1.4	20.1	100.8 ± 0.6	2.62	3.24
Philozac (Amoun, Egypt)	20	18.9	99.2 ± 0.8	19.8	99.07 ± 1.7	1.38	6.87
Flutin (Eipico, Egypt)	20	18.8	102.0 ± 1.8	19.8	99.4 ± 0.9	2.89	4.54
Depreban (Amirya, Egypt)	20	20.7	102.6 ± 1.5	19.4	97.2 ± 0.8	1.45	2.07

^a^ Mean of three replicates. ^b^
*t*-Student and *F*-test test at 95% confidence level values are 4.30 and 19.00, respectively.

**Table 7 membranes-12-00446-t007:** Determination of FLX in different pharmaceutical preparations using ionophore (II) membrane-based sensor.

Pharmaceutical Product and Source	Nominal Content Taken, mg/Tablet	Found, mg/Tablet				*t*-Student Test ^b^	*F*-Test
		Proposed Method	Mean ^a^ (%) ± SD	Reference Method	Mean ^a^ (%) ± SD		
Prozac (Lilly, France)	20	19.7	98.5 ± 0.4	20.1	100.8 ± 0.6	2.85	3.12
Philozac (Amoun, Egypt)	20	20.9	104.5 ± 0.5	19.8	99.07 ± 1.7	3.138	5.24
Flutin (Eipico, Egypt)	20	19.3	96.5 ± 0.8	19.8	99.4 ± 0.9	2.93	3.37
Depreban (Amirya, Egypt)	20	19.5	97.5 ± 0.4	19.4	97.2 ± 0.8	2.34	2.16

^a^ Mean of three replicates. ^b^
*t*-Student and *F*-test test at 95% confidence level values are 4.30 and 19.00, respectively.

**Table 8 membranes-12-00446-t008:** Potentiometric assessment of FLX in different spiked serum samples.

Sample No.	Amount of FLX Added, μM	Ionophore I		Ionophore II		Ionophore III	
		Amount of FLX Found, μM ^a^	Recovery, %	Amount of FLX Found, μM ^a^	Recovery, %	Amount of FLX Found, μM ^a^	Recovery, %
1	8.0	7.8 ± 0.8	97.5	7.7 ± 0.9	96.3	7.8 ± 0.8	97.5
2	10.0	9.7 ± 0.6	97.0	9.5 ± 0.4	95.0	9.8 ± 0.4	98.0
3	15.0	15.5 ± 0.2	103.3	15.1 ± 0.3	100.6	14.8 ± 0.3	98.6
4	20.0	19.6 ± 0.7	98.0	20.1 ± 0.6	100.5	19.8 ± 0.1	99.0

^a^ Mean of three replicates.

## Data Availability

The data presented in this study are available on request from the corresponding author.

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
