# Peer review of "All-Solid-State Potentiometric Platforms Modified with a Multi-Walled Carbon Nanotubes for Fluoxetine Determination"

_membranes, 2022, doi:10.3390/membranes12050446_

Round 1

Reviewer 1 Report

In the manuscript, cost-effective screen-printed potentiometric platforms for simple and accurate assessment of Fluoxetine (FLX) drug have designed and characterized.  Multiwalled carbon nanotubes have been used in the manuscript as a lipophilic  ion-to-electron transducing material and sodium tetrakis [3,5-bis(trifluoromethyl)phenyl] borate (NaTFPB) has been used as an anionic excluder. The sensors have been introduced for successful determination of FLX drug in different pharmaceutical dosage forms. The idea behind this research work is original. After rational reply to the following raised issues (minor revision).  The paper is suitable to be published in the journal after the following minor revisions have been made. 

1-)  Similar articles should be cited before the Table 1  These materials and comparison of the sensors with potentiometric ion-selective electrodes (ISEs) are not new.

2-) Some typing and english grammar errors should be corrected for in the abstract and introduction section.  Similar typing errors should be checked the other sections.

3-) The authors should stated in the text why they have introduced  four commercially available drugs containing FLX has been chosen to test the applicability of the presented electrodes.

4-) The definition of the parameters should be written before tables and  figures. The authors ahould check the whole manuscript about the definition of parameters.

5-) There are various abbreviation errors in general in the text. Abbreviations should be defined at least once at the first occurrence and then used in abbreviated form. The authors should check the all abbreviation.

6-) There are some misunderstanding table and figure names. The authors should check the all names of figures and tables. The readers can not follow in this situation.

7-) The authors should check the manuscript against badly constructed sentences and grammatical errors.

8-) Referances should be enriched. Following articles about the eigen value analysis buckling or vibrational response can be added in the introduction section.

  • Axial vibration analysis of a Rayleigh nanorod with deformable boundaries. Microsystem Technologies26(8), 2661-2671.

Author Response

Reply to reviewers’ comments

 1-) Similar articles should be cited before the Table 1.  These materials and comparison of the sensors with potentiometric ion-selective electrodes (ISEs) are not new.

Done.

2-) Some typing and English grammar errors should be corrected for in the abstract and introduction section.  Similar typing errors should be checked the other sections.

English language is revised by a native English speaker.

3-) The authors should stated in the text why they have introduced four commercially available drugs containing FLX has been chosen to test the applicability of the presented electrodes.

It is clarified in Section 2.5.

4-) The definition of the parameters should be written before tables and figures. The authors should check the whole manuscript about the definition of parameters.

Done. All parameters are written before tables and figures.

5-) There are various abbreviation errors in general in the text. Abbreviations should be defined at least once at the first occurrence and then used in abbreviated form. The authors should check all abbreviation.

All abbr. are corrected and identified.

6-) There are some misunderstanding table and figure names. The authors should check the all names of figures and tables. The readers can not follow in this situation.

Done and corrected.

7-) The authors should check the manuscript against badly constructed sentences and grammatical errors.

English language is revised by a native English speaker.

8-) References should be enriched. Following articles about the eigen value analysis buckling or vibrational response can be added in the introduction section.

  • Axial vibration analysis of a Rayleigh nanorod with deformable boundaries. Microsystem Technologies, 26(8), 2661-2671.

48 references are presented in the manuscript. Almost of these references are updated and have a high eigen factor.

Reviewer 2

The authors presented a paper that describes the design and fabrication of a MWCNT based sensor for the detection of FLX. Results confirmed successful detection with detection limits which are comparable to other FLX electrochemical sensors. 

There are a number of grammatical and typing errors.

English language is revised by a native English speaker.

Reviewer 2 Report

The authors presented a paper that describes the design and fabrication of a MWCNT based sensor for the detection of FLX. Results confirmed successful detection with detection limits which are comparable to other FLX electrochemical sensors. 

There are a number of grammatical and typing errors.

Author Response

(The authors gave the same response as above.)
